# Molecular Modification Enhances Xylose Uptake by the Sugar Transporter KM_SUT5 of *Kluyveromyces marxianus*

**DOI:** 10.3390/ijms25158322

**Published:** 2024-07-30

**Authors:** Xiuyuan Luo, Xi Tao, Guangyao Ran, Yuanzhen Deng, Huanyuan Wang, Liyan Tan, Zongwen Pang

**Affiliations:** College of Life Science and Technology, Guangxi University, Nanning 530004, China; xiuyuan0607@126.com (X.L.);

**Keywords:** *Kluyveromyces marxianus*, sugar transporter, point mutation, carboxyl terminus truncation, D-xylose

## Abstract

This research cloned and expressed the sugar transporter gene KM_SUT5 from *Kluyveromyces marxianus* GX-UN120, which displayed remarkable sugar transportation capabilities, including pentose sugars. To investigate the impact of point mutations on xylose transport capacity, we selected four sites, predicted the suitable amino acid sites by molecular docking, and altered their codons to construct the corresponding mutants, Q74D, Y195K, S460H, and Q464F, respectively. Furthermore, we conducted site-directed truncation on six sites of KM_SUT5p. The molecular modification resulted in significant changes in mutant growth and the D-xylose transport rate. Specifically, the S460H mutant exhibited a higher growth rate and demonstrated excellent performance across 20 g L^−1^ xylose, achieving the highest xylose accumulation under xylose conditions (49.94 μmol h^−1^ gDCW-1, DCW mean dry cell weight). Notably, mutant delA554-, in which the transporter protein SUT5 is truncated at position delA554-, significantly increased growth rates in both D-xylose and D-glucose substrates. These findings offer valuable insights into potential modifications of other sugar transporters and contribute to a deeper understanding of the C-terminal function of sugar transporters.

## 1. Introduction

Lignocellulosic materials have emerged as a promising alternative for second-generation (2G) fuel ethanol production, providing a viable solution to meet increasing energy demands [1,2]. In addition to glucose, which serves as the primary constituent, lignocellulosic materials also contain approximately 30% pentose sugars, predominantly D-xylose and L-arabinose [3]. Efficient co-fermentation of both glucose and pentose sugars is crucial for achieving optimal 2G fuel ethanol production [4,5]. The initial step in carbohydrate metabolism involves sugar uptake, wherein the capacity of sugar transporters directly influences cellular metabolic rates [6].

Traditional *Saccharomyces cerevisiae* strains lack specific pentose transporters, and their hexose transporters, such as Hxt7p or Gal2p, exhibit significantly higher affinity towards glucose than xylose [7]. The limited rate of pentose transport represents a significant bottleneck that restricts the utilization of D-xylose [8,9]. Consequently, extensive research has been devoted to identifying efficient D-xylose and/or L-arabinose transporters in other D-xylose-utilizing strains with the aim of enhancing *S. cerevisiae* pentose transport capacity through heterologous expression. Examples of such transporters include Gxf1p and Gxs1p from *Candida intermedia*; XltA, XltB, and XltC from *Aspergillus niger*; An25p from *Neurospora crassa*; and Lat1p and Lat2p from *Ambrosiozyma monospora*, among others [10,11,12,13]. Despite the discovery of numerous novel transporters, the achieved xylose transport rates by these carriers remain unsatisfactory.

Various strategies, such as genetic engineering and evolutionary engineering, have been employed to enhance xylose transport performance [14,15,16]. The mutant Gxs1p (gxs1-2.3) exhibited a 70% increase in growth rate on D-xylose compared to the wild-type Gxs1p in recombinant hxt null (lacking all hexose transporter proteins) *S. cerevisiae* strains [17,18]. An evolved strain identified a fusion protein, Hxt36, derived from endogenous transporters Hxt3p and Hxt6p that enabled co-transport of D-xylose and glucose [16]. Additionally, a mutant of *S. cerevisiae* endogenous transporter HXT2p (C505P) was discovered that significantly enhanced D-xylose transport preference [14]. Carboxyl terminal truncation achieved through random mutagenesis notably improved the relative growth rate, but the exact function of the C-terminal fragment remains unclear [19]. Therefore, there is an urgent need to discover or modify efficient transporters that can specifically and efficiently transport pentose sugars without being inhibited by glucose during sugar co-fermentation processes.

*Kluyveromyces marxianus*, a thermotolerant yeast capable of fermenting various sugars including glucose, fructose, xylose, and arabinose, represents a valuable resource for acquiring pentose transporter genes [20,21]. In this study, our focus was on the natural D-xylose-utilizing strain *K. marxianus* GX-UN120 to clone a novel pentose transporter. By conducting whole-genome analysis of *K. marxianus* DMKU 3-1402 [22], we identified 28 sugar transporter proteins in *K. marxianus* GX-UN120, among which the sugar transporter protein KM_SUT5 was associated with a putative transporter gene, KLMA_20830 (accession: XM_022818478), showing high homology. During the investigation of its transport capacity, we discovered that KM_SUT5 exhibits excellent transportation ability for xylose, which motivated us to further study it. To gain insights into its functional characteristics, we conducted investigations using the hxt null *S. cerevisiae* strain EBY-XYL. I-TASSER modeling [23,24,25] to examine the structural features of KM_SUT5p along with other xylose transporters revealed significant structural similarity within their 12 transmembrane domains while displaying limited similarity in the C-terminal region located after TM12. The key amino acid sites within both transmembrane domains and the C-terminal region play crucial roles in determining the functionality and stability of these transporter proteins [26,27].

To further investigate the functional impact of the C-terminal region and specific amino acid sites on xylose transport capacity, we conducted molecular modifications on KM_SUT5. The transport capacity of the transporter protein can be influenced by both point mutations and C-terminal truncations. These findings present an intriguing avenue for sugar transporter engineering and suggest that further research in this direction could yield valuable insights into enhancing xylose transport capacity and overcoming glucose inhibition in diverse biotechnological applications.

## 2. Results

### 2.1. The Subcellular Localization of KM_SUT5p

The KM_SUT5 gene was genetically modified by fusing the GFP gene at its C-terminus, and this fusion construct was subsequently introduced into *S. cerevisiae* EBY-XYL to investigate the subcellular localization of the resulting fusion protein. Visualization of the expressed fusion protein was achieved through fluorescence imaging, revealing predominant green fluorescence signal localization in the cell membrane (Figure 1). Additionally, some fluorescence signal was observed in the cytoplasm. This observation suggests that during GFP production, prior to complete membrane localization, it may exhibit transient cytoplasmic fluorescence [27]. Collectively, these findings provide compelling evidence supporting KM_SUT5p as a transmembrane protein correctly localized to the cell membrane.

### 2.2. The Symporter Assay of KM_SUT5p

Monosaccharide transporters can be classified into two distinct types: facilitators and symporters. Facilitators are known for their high efficiency in taking up small molecules without the consumption of energy, although they have relatively low uptake capacities. On the other hand, symporters exhibit higher affinities for small molecules but lower translocation efficiencies compared to facilitators. They require energy and are coupled with H^+^ ions, resulting in a significant increase in extracellular pH.

To assign the transporter type to the putative transporter under investigation, sugar–proton symport experiments were conducted [28], using *S. cerevisiae* EBY-P as a control. The pH changes were measured upon the addition of a D-xylose solution. Remarkably, strain *S. cerevisiae* EBY-SUT5 exhibited a significant increase in pH, while the control group EBY-P did not demonstrate such an effect (Figure 2). These findings strongly support that KM_SUT5p functions as a D-xylose symporter, providing an explanation for the observed growth in the presence of D-xylose.

### 2.3. The Characteristics of KM_SUT5p for Sugar Transportation in EBY-XYL

The sugar transport capacities of KM_SUT5p were evaluated using a drop-test assay to determine their potential to confer the ability of *S. cerevisiae* EBY-XYL to utilize diverse sugars. In Figure 3A, distinct growth patterns were observed for *S. cerevisiae* EBY-SUT5 compared to the control strain, *S. cerevisiae* EBY-P. Specifically, *S. cerevisiae* EBY-SUT5 exhibited restored growth on sorbose and displayed limited growth on galactose. However, it was unable to support growth on other hexose sugars such as glucose, mannose, and fructose. Interestingly, noteworthy growth was observed for both xylose and ribose in *S. cerevisiae* EBY-SUT5.

To gain a more comprehensive understanding of the transport capacity of KM_SUT5p, growth curves were obtained through liquid cultures, as shown in Figure 3B. Notably, the strain exhibited superior growth in the presence of sorbose, reaching an OD_600_ value of 0.46. In xylose and ribose media, the corresponding OD_600_ values were 0.38 and 0.31, respectively. However, when grown in glucose medium, the maximum growth only reached an OD_600_ of 0.19, representing merely half of that observed in xylose medium. Furthermore, the strain displayed maximum growth rates of 0.032 ± 0.001 h^−1^ and 0.061 ± 0.007 h^−1^ in glucose and xylose media, respectively, indicating a higher transport capacity for xylose compared to glucose by KM_SUT5p while efficiently transporting all three pentose sugars overall. Conversely, no significant growth was observed when mannose or fructose was used as carbon sources, consistent with results from drop-test assays, suggesting that KM_SUT5p lacks the ability to transport mannose and fructose.

### 2.4. Effect of the Mutations in the Amino Acid Site of KM_SUT5p on the Affinity for Xylose

Based on the results of molecular docking, the docking of SUT5 protein and xylose ligand can be observed in close proximity to protein amino acid residues Q74, Y195, S460, and Q464. These residues are likely to play a crucial role in substrate recognition during xylose transport (Figure 4A). Additionally, the docking of control XylE (D-xylulose-plasmonic symporter protein from *Escherichia coli*) with small molecules of xylose was also examined. Appendix A illustrates the alterations in binding energy of each mutant protein to xylose by manually modifying the remaining 19 amino acids’ sequence, constructing 3D models for each mutant through homology modeling, and subsequently conducting docking simulations with the D-xylose molecule using AutoDock version 1.5.6.

In a solid medium containing 20 g L^−1^ xylose, mutant strains Q74D, Y195K, and S460H exhibited enhanced growth compared to the original strain; however, no growth was observed in the Q464F mutant (Figure 4B). When grown in a solid medium with 1 g L^−1^ xylose, the original strain demonstrated better growth than in 20 g L^−1^ xylose, indicating potentially higher transport efficiency of KM_SUT5 at lower concentrations of xylose. Interestingly, the mutant strains Q74D, Y195K, and S460H displayed similar growth patterns to the original strain under low xylose concentrations, suggesting that mutations at these potential sites did not enhance the sugar transporter’s transport capacity. In liquid medium containing 20 g L^−1^ xylose, significant growth improvements were observed for mutant strains Q74D (45.3%), Y195K (15.7%), and S460H (52.4%) compared to the original strain, whereas the Q464F mutant showed diminished growth trend in liquid culture consistent with plate experiments. The growth of all recombinant strains was observed to be lower in the medium containing 1 g L^−1^ xylose compared to 20 g L^−1^ xylose (Figure 4C). However, the S460H mutant, which exhibited robust growth in the presence of 20 g L^−1^ xylose, still demonstrated a significant improvement in cultures with low xylose concentrations, showing an approximate enhancement of 61.2%. Docking results revealed that the S460 site formed a pair of hydrogen bonds with xylose molecules. Substituting it with histidine, which possesses two nitrogen atoms, facilitated the formation of hydrogen bonds with xylose molecules. Furthermore, due to its hydrophobic spatial arrangement within the molecule, this mutation may contribute to enhanced passage of xylose molecules. These findings suggest that Y195K and S460H mutations enhance xylose transport, while Q464F mutation diminishes it.

### 2.5. Effect of the C-Terminal Truncation of KM_SUT5p on the Affinity for Xylose

To address this issue, we performed a comparative analysis of the secondary structure of multiple C-terminals derived from xylose transporters (Figure 5A). The results revealed a relatively low sequence similarity among these segments, with only 25.3% identity observed. Notably, the length of these terminals ranged from 50 to 70 amino acid residues, exhibiting minimal variation. Furthermore, we identified a conserved sequence motif, KG-XX-LEE, located at the N-terminal region of the C-terminal segment and forming an α-helix conformation. However, no discernible structural or sequence similarities were observed in the fragments following this conserved region. In order to investigate the influence of both C-terminal length and the α-helical motif within the conserved region on KM_SUT5p sugar transporter function, we selected six specific sites for truncation (indicated by black arrows in Figure 5A), namely, delL509-, delP519-, delV529-, delV534-, delK544-, and delA554-.

We assessed the growth of recombinant strains in 2% xylose and 0.1% xylose (Figure 5B,C). In the presence of 2% xylose, the strain with truncation at delP519- exhibited the highest growth increase of 32%, followed by delV534- with a 21% increase, delK544- with a 27% increase, and delA554- with a 29% increase. The delL509- mutant did not show a significant growth improvement, while delV529- displayed lower growth compared to the original strain. The original strain exhibited a growth rate of only 60% under conditions of 0.1% xylose, whereas its growth rate increased to 82% in the presence of 2% xylose. These findings suggest that KM_SUT5p functions as a symporter-type transporter maintaining relatively stable transport efficiency under low concentrations of xylose. All other mutants exhibited enhanced maximum growth at low levels of xylose concentration. The delP519- mutant exhibited a remarkable 84% enhancement in maximum growth value (OD_600_ = 0.427) compared to the original strain, while the delK544- mutant demonstrated a substantial improvement with an OD_600_ of 0.419, representing an impressive 81% increase over the original strain.

### 2.6. Effect of Mutations in the Amino Acid Site and C-Terminal Truncation of KM_SUT5p on D-Xylose Transport

In liquid cultures containing mixed sugars (Figure 6A,B), all three mutant strains at the amino acid sites exhibited significant growth except for Q464F, while all C-terminal truncation mutant strains showed significantly enhanced growth. S460H and delA554- displayed growth rates of 0.064 ± 0.002 h^−1^ and 0.081 ± 0.002 h^−1^, respectively, representing a 100.05% and 153.1% increase compared to the original strain.

Furthermore, we quantified the xylose transport rate of all mutant strains under 2% xylose and mixed sugar conditions (Figure 6C,D). At a concentration of 20 g L^−1^ xylose, S460H and Q74D exhibited significantly higher intracellular xylose contents of 49.94 μmol h^−1^ gDCW^−1^ and 42.78 μmol h^−1^ gDCW^−1^, respectively, compared to the original strain EBY-SUT5 which only had a content of 15.78 μmol h^−1^ gDCW^−1^. In the presence of mixed sugars, the intracellular xylose content in S460H and Q74D was approximately half that observed in pure xylose conditions at 29.95 μmol h^−1^ gDCW^−1^ and 29.49 μmol h^−1^ gDCW^−1^, respectively; however, it still showed an almost three-fold increase compared to EBY-SUT5. These findings suggest that mutations in S460H and Q74D enhance the capacity for xylose transport by the sugar transporter KM_SUT5 while also improving its performance in transporting xylose within mixed-sugar environments despite incomplete elimination of glucose inhibition.

Regarding the C-terminal truncation mutants, delA554- and delP519- exhibited significantly higher intracellular xylose contents of 43.25 μmol h^−1^ gDCW^−1^ and 32.27 μmol h^−1^ gDCW^−1^, respectively, which were 174.1% and 104.5% higher than the original strain EBY-SUT5, respectively. However, under mixed-sugar conditions, all mutants showed a significantly lower intracellular xylose content compared to that in pure xylose, indicating the inhibitory effect of glucose on xylose transport. Among them, delA554- demonstrated a substantial decrease in intracellular xylose content to 10.76 μmol h^−1^ gDCW^−1^, which was 75.1% lower than that in xylose, while delP519- maintained the highest intracellular xylose content under mixed-sugar treatment despite a reduction of 52.9% compared to that in xylose. Furthermore, when incubated with either pure xylose or mixed-sugar conditions, the delL509- strain only exhibited modest increases in intracellular xylose content by 44.2% and 69.9%, respectively, suggesting that truncation at this site can effectively alleviate glucose inhibition.

## 3. Discussion

The substitution of lysine for the phenyl ring at Y195 significantly impacts the glucose transport process of the KM_SUT5 protein. It markedly enhances strain growth under mixed-sugar conditions and reduces glucose inhibition during xylose transportation. Wang et al.’s study proposed that long-chain or non-polar amino acids, such as phenylalanine and lysine, located within the channel, may induce steric repulsion and impede glucose entry [29]. This finding could elucidate the functional alterations observed with the Y195K mutation. Amongst all mutations, S460 site alteration exhibits the most pronounced improvement in xylose transport capacity. SUT5-S460H demonstrates substantially enhanced growth and xylose accumulation, with a growth rate 1.5 times higher than that of the EBY-SUT5 starting strain and a 2.1-fold increase in xylose absorption. According to 3D model prediction, the S460 site is positioned inside the protein channel at 1.9 Å from the xylose molecule, enabling stable hydrogen bonding formation with it. The substitution of histidine residue abolishes the hydrophobic cavity within the protein and promotes hydrogen bonding with xylose molecules in comparison to serine, potentially accounting for the notable enhancement in xylose transportation observed with S460H mutation. Conversely, the Q464F mutation severely impairs KM_SUT5’s ability to transport both xylose and glucose. Previous studies have suggested that glutamine plays a pivotal role in substrate recognition among sugar transporters, which could explain the diminished functionality associated with this mutation [30,31]. Furthermore, given that the Q464 site is situated at the outer periphery of KM_SUT5’s protein channel alongside phenylalanine at position 465, it introduces a phenylalanine substitution at Q464 that increases steric hindrance at the channel entrance, rendering it arduous for sugar molecules to enter. Consequently, this alteration results in significantly reduced transport capacity.

In terms of structural characteristics, it is observed that delP519- retains the conserved secondary structure KG-XX-LEE, while delL509- represents a complete truncation of the C-terminal region. Notably, delP519- significantly enhances xylose transport capacity in the sugar transporter KM_SUT5p, suggesting that preservation of this conserved secondary structure confers benefits for xylose transport. This conserved α-helix at the C-terminal is also present in XylE, a xylose-specific sugar transporter from bacteria. Moreover, mutations delV534- and delA554- exhibit significant differences in their ability to transport xylose after the delP519- site; notably, delA554- surpasses delP519- but is severely inhibited by glucose. These findings indicate that truncation length may not directly correlate with sugar transporter performance and that preserving conserved α-helix segments may not always represent an optimal truncation site. However, such preservation could be advantageous for enhancing xylose transport capacity. Additionally, previous studies have demonstrated that alterations in the C-terminus impact protein stability and potentially contribute to reduced xylose transport capacity [26,27]. Furthermore, these C-terminal truncations also improve the transporter’s ability to efficiently transport low concentrations of both xylose and glucose by fine-tuning the overall protein structure following truncation [32].

Among these sites, S460H significantly enhances xylose transport capacity, while Y195K greatly improves glucose transport capacity. Conversely, the Q464F mutation results in an almost complete loss of sugar transport capacity. Additionally, truncation mutants exhibit improved growth under low D-xylose concentrations. The relationship between C-terminal length and function does not appear to be direct; further investigation is required to uncover underlying reasons for this change. These findings underscore the crucial role of the C-terminal in xylose transport and provide a new direction for molecular modifications of sugar transporters.

## 4. Materials and Methods

### 4.1. Plasmid and Strains

In our experimental setup, the plasmid pRS424-Hxt7 was utilized as the vector for expressing the putative transporter. This plasmid contained the Hxt7p promoter, Hxt7t terminator, and TRP1 selection marker gene [33]. The open reading frame (ORF) encoding the transporters and their mutants was inserted into the corresponding restriction enzyme sites between Hxt7p and Hxt7t on the pRS424-Hxt7 plasmid using a reverse polymerase chain reaction (PCR) strategy for site-directed mutagenesis. To fuse the KM_SUT5 gene fragment into the plasmid pRS424-Hxt7-GFP, resulting in pRS424-Hxt7-SUT5-GFP, a fusion technique was employed. To establish the basic D-xylose metabolic pathway in hxt null background strains, we constructed plasmid YEplac195-XYL1-XYL2 by inserting D-xylose reductase gene XYL1 (UniprotKB number: P31867) and D-xylitol dehydrogenase gene XYL2 (UniprotKB number: P22144) from *Scheffersomyces stipitis* into linearized YEplac195 using the DNA assembler method [34,35]. All plasmids were chemically transformed into *Escherichia coli* DH5α competent cells to construct the plasmid library. The *S. cerevisiae* EBY-XYL strain was derived from EBY.VW4000 [18], which lacked all 18 natural hexose transporters but complemented the xylose metabolism pathway through electro-transformation of plasmid YEplac195-XYL1-XYL2. For expression in *S. cerevisiae* EBY-XYL, yeast cells were electroporated with specific parameters: voltage of 1.65 kV, capacitor of 25 μF, resistance value of 200 Ω, and a shock time of 5 ms. In this experimental setup, maltose was utilized as the carbon source for the culture. To prepare competent cells, *S. cerevisiae* cells were cultured until they reached the late-log phase (1 × 10^7^ cells mL^−1^). The cells were then harvested, suspended in a buffer solution (containing 25 mM dithiothreitol, 1 M sorbitol, and 20 mM Hepes at pH 7.5) at a temperature of 30 °C for a duration of 15 min and subsequently subjected to three rounds of ice-cold sterilized 1 M sorbitol washes through centrifugation (8000 rpm for 6 min). For reference purposes, Appendix A provides a comprehensive list of the plasmids and *S. cerevisiae* strains employed in this study. Furthermore, Appendix A summarizes the primers used along with their corresponding restriction enzyme sites for plasmid construction. Sequence data supporting the findings presented in this study have been deposited in GenBank under accession number OL743188.

### 4.2. Cultivation Conditions and Growth Test

The *K. marxianus* GX-UN120 strain, obtained from a previous study [36], was maintained on YPD liquid medium (10 g L^−1^ yeast extract, 20 g L^−1^ peptone, and 20 g L^−1^ glucose) and used as the source for obtaining sugar transporter genes. *E. coli* DH5α was employed for plasmid propagation and amplification, cultured in LB liquid medium (10 g L^−1^ peptone, 5 g L^−1^ yeast extract, and 10 g L^−1^ NaCl) with 100 μg mL^−1^ ampicillin. *S. cerevisiae* EBY-XYL was cultivated in YPM liquid medium (20 g L^−1^ peptone, 10 g L^−1^ yeast extract, and 20 g L^−1^ maltose) for maintenance purposes. Synthetic Complete (SC) medium (Trp Ura Minus Media), supplemented with either one or two tested sugars at a concentration of 20 g L^−1^ each, was used to culture strains containing the plasmids based on specific experimental requirements.

For spotting assays, cells were harvested from a 16 h culture in YPM medium with maltose, washed, and resuspended in 0.9% NaCl. After incubating for 9 h at 30 °C, the cell density was normalized to an OD_600_ of 1.0. Subsequently, 2 μL of serially diluted cells (10-fold dilution) was spotted onto test plates and cultivated at 30 °C. The growth status was evaluated after a period of 5 days.

To prepare for transport rate and symporter assays, strains were initially cultured in YPM or SC medium with maltose for a duration of 16 h. Then, they were transferred to fresh medium with an initial OD_600_ of 0.1 and cultured for an additional period of 16 h. Pre-cultured strains were collected, washed twice, and utilized as seed cells for the respective tests.

The growth status in the liquid medium was determined by quantifying the culture’s optical density at 600 nm using a UV–vis spectrophotometer (UVmini1240, Shimadzu, Kyoto, Japan). The maximum growth rate was calculated from the growth curve to compare growth capacities. Dry cell weight (DCW) was estimated using the formula DCW (mg mL^−1^) = 0.2365 × OD_600_ + 0.1149 [9].

### 4.3. Fluorescence Localization

To investigate the subcellular localization of the putative transporter in *S. cerevisiae* EBY-XYL, we conducted fluorescence localization analysis using GFP. The putative transporter genes were fused with GFP at the C-terminus and introduced into *S. cerevisiae* via electroporation. Subsequently, the resulting strains carrying the GFP-fusion plasmids were cultured in SC liquid medium supplemented with 2% maltose at 30 °C. Cells in the exponential phase were harvested, washed, and resuspended in phosphate buffered saline (PBS) with a pH of 6.5 (50 mmol L^−1^). Fluorescence images were captured utilizing a confocal laser scanning microscope (MRC-1024, Bio-Rad, Hercules, CA, USA).

### 4.4. Symporter Assay

To determine the transporter type, we performed a symporter assay by measuring pH changes using a pH meter (LICHEN, pH-100, Shanghai, China). A 100 mL fresh culture was collected after 16 h of growth and subsequently washed three times with ice-cold sterile water. The cells were then resuspended in sterile water to achieve an OD_600_ of approximately 20. The pH of the resuspended cells (20 mL) was adjusted to 5.0 using HCl. Following that, a 400 μL sugar solution (D-xylose or D-glucose, 20 g L^−1^) was added. Symporter transporters coupled with protons during sugar uptake, resulting in an elevation in pH levels; however, facilitator transporters did not exhibit any significant change in pH levels [28].

### 4.5. Molecular Docking

The KM_SUT5p model and other models were generated using the I-TASSER online tool (https://zhanggroup.org/I-TASSER/, accessed on 6 March 2023), which employs advanced algorithms for protein structure prediction. The predicted protein’s localization information was obtained from UniProt (https://www.uniprot.org/, accessed on 6 March 2023), a comprehensive database for protein sequence and functional information. The D-xylose molecule (access number: 1529215) used in the docking analysis was acquired from the ZINC online website (https://zinc.docking.org/, accessed on 6 March 2023), which provides a collection of commercially available compounds for virtual screening. For the docking analysis of KM_SUT5 with xylose molecules, AutoDock software (Ver. 4.2.6) was utilized. The AutoGrid area dimensions were set as 80 (x), 40 (y), and 40 (z), with a grid spacing of 0.375 units. Docking calculations employed the Lamarckian genetic algorithm (LGA), a widely used approach for protein–ligand docking [37]. The docking method used was semi-flexible docking. Docking results were ranked based on binding energy, where higher affinity indicates stronger interaction between substrate and protein. To visualize the binding model of the KM_SUT5 protein with D-xylose molecules, PyMOL protein model visualization software was employed to comprehensively analyze docking results and visualize protein–ligand interactions [38].

### 4.6. Molecular Modification

The site-directed mutagenesis of KM_SUT5p was performed using the Mut Express MultiS Fast Mutagenesis Kit (V2-C215, Vazyme, Nanjing, China), specifically designed for efficient and reliable introduction of targeted mutations. Initially, inverse PCR was conducted with plasmid DNA as the template along with mutation primers (refer to Appendix A for details). Based on molecular docking results, individual binding sites were substituted with the remaining 19 amino acids, and homology modeling was employed to construct 3D models of each mutant. Subsequently, Autodock software was utilized to perform docking simulations between each mutant protein and D-xylose molecule. The lowest binding energy changes of the mutant proteins with xylose are depicted in Appendix A. Selected amino acid residues in KM_SUT5p were replaced by codons corresponding to the desired target amino acid or stop codon (TAA or TAG). Following PCR amplification, resulting products were treated with Dpn I enzyme that specifically cleaves methylated and hemimethylated DNA. This step eliminates template plasmid DNA while retaining newly synthesized mutated DNA fragments. Circularization of PCR products was achieved using EXnase II ligase, which efficiently seals nicks in the DNA backbone. To confirm successful introduction of mutations, the resulting mutation plasmids underwent sequencing analysis, ensuring the accuracy of the desired mutations. Once verified, plasmids were transformed into *E. coli* for propagation and amplification.

Finally, the validated mutation vectors were subsequently introduced into yeast cells to generate the respective mutated yeast strains, thereby facilitating the functional characterization of the mutant transporters.

### 4.7. Transport Rate of D-Xylose

The *S. cerevisiae* EBY-SUT5 and its mutant strains in logarithmic phase were harvested and washed three times with sterile water to eliminate any extraneous substances. Subsequently, the cells were resuspended in sterile water and incubated for a period of 6–8 h to ensure an adequate recovery period. Following this incubation, an equal volume of a 4% xylose solution or a mixed-sugar solution was added to the cell suspension, followed by a 1-h incubation period. To ensure the removal of any external sugars, the cells were then washed three times with sterile water. For determining the D-xylose transport rate, appropriate amounts of glass beads were added to the collected organisms, and the cells were shaken in a cell crusher at 60 Hz for 30 s; this shaking process was repeated ten times to fully disrupt the cells. High-performance liquid chromatography (HPLC) (e2695, Waters, Milford, MA, USA) coupled with pre-column derivatization using 1-phenyl-3-methyl-5-pyrazolone (PMP) was employed for analyzing D-xylose content [39]. The HPLC system utilized a Symmetry^®^ C18 column (4.6 mm × 250 mm, 5 μm particle size). The mobile phase consisted of acetonitrile and sodium acetate buffer (pH 6.5) at a ratio of 19:81 with a concentration of 0.1 mol/L, respectively. The detection wavelength was set at 250 nm while maintaining column temperature at 40 °C and a flow rate at 1 mL min^−1^. The D-xylose accumulation rate expressed as µmol h^−1^ gDCW^−1^ (micromoles per hour per gram dry cell weight), was calculated based on HPLC analysis results. This quantitative measurement provides valuable insights into D-xylose transport efficiency by investigated strains or mutants. 

## Figures and Tables

**Figure 1 ijms-25-08322-f001:**
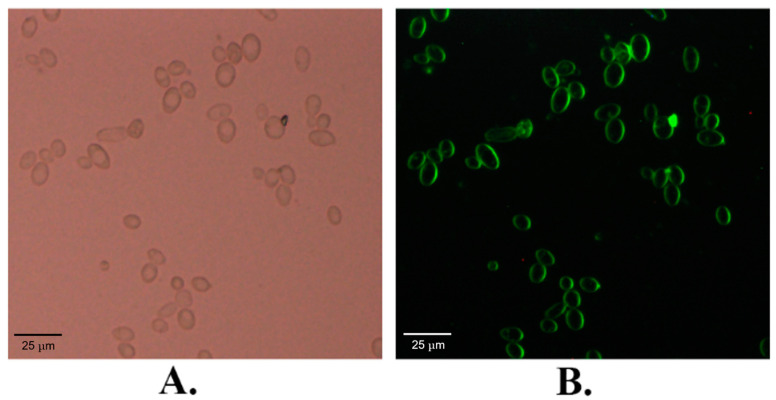
Subcellular localization based on sugar transporter GFP fluorescence. GFP was fused at the C-terminal of the transporter and was expressed in genetically modified *S. cerevisiae* EBY-XYL. (**A**) Bright field, (**B**) fluorescence.

**Figure 2 ijms-25-08322-f002:**
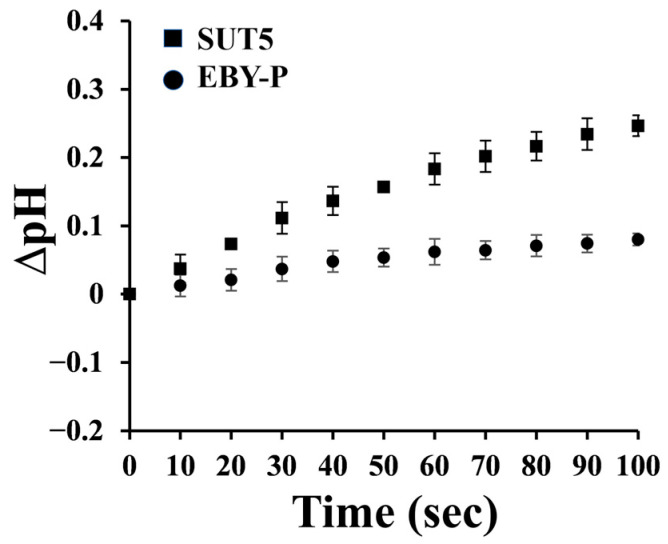
The change in pH value during the transport process indicated the transport type. Zero seconds was the time when D-xylose added. The ΔpH were the differences between the initial pH 5.0 and the recorded value. The notable increase suggests that the transporter is an H^+^ symporter. The error bars represent the standard deviation of the biological triplicates.

**Figure 3 ijms-25-08322-f003:**
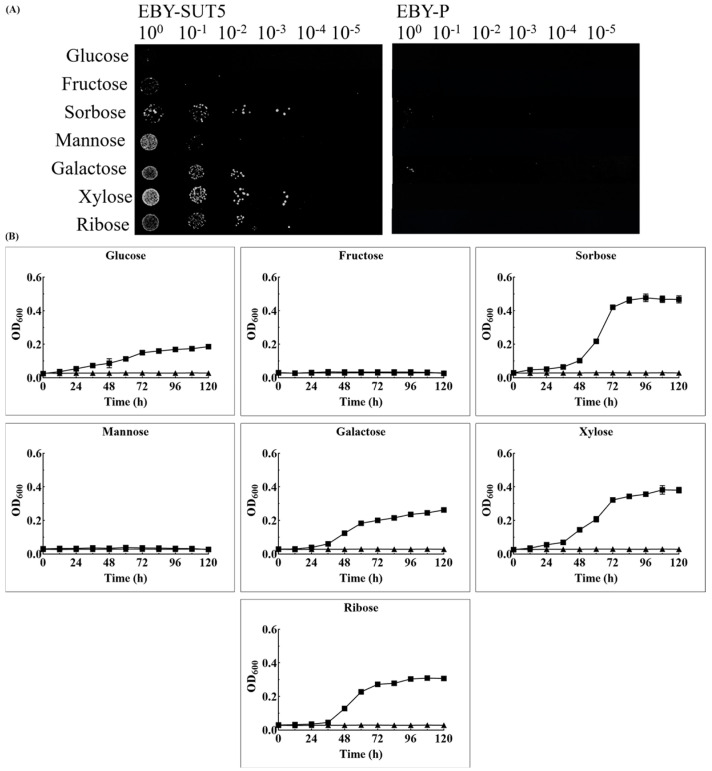
The ability to transport sugar of *S. cerevisiae* EBY-SUT5. Recombinant *S. cerevisiae* expressing KM_SUT5 was harvested in log stage, starvation treated, and adjusted to OD_600_ 1.0. Suspension onto (**A**) solid or (**B**) liquid SC-TRP-URA medium containing 2% sugars for 5 days, triangle for EBY-P, rectangle for EBY-SUT5. Error bars represent the standard deviation of the biological triplicates.

**Figure 4 ijms-25-08322-f004:**
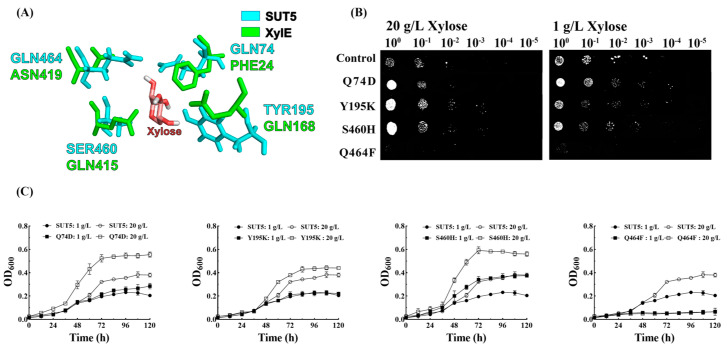
Molecular docking results of KM_SUT5 and growth of point mutant strains under different concentrations of xylose. (**A**) Comparison of residues bound to xylose in XylE and SUT5; cyan color indicates SUT5, green indicates XylE. Growth of point mutant strains in (**B**) solid and (**C**) liquid synthetic medium containing 2% or 0.1% xylose for 5 days. Error bars represent the standard deviation of the biological triplicates.

**Figure 5 ijms-25-08322-f005:**
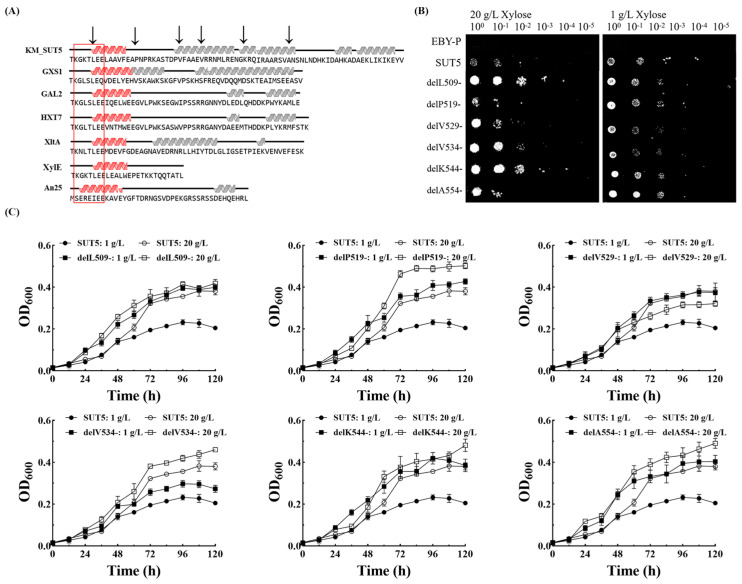
Construction of carboxy-terminal truncation mutant strains and their growth under different concentrations of xylose. (**A**) Comparison of C-terminal sequence and secondary structure. The helical pattern represents the α-helical, the red parts indicate similar structures, and the black arrows indicate truncated sites, GXS1 from *C. intermedia*, GAL2 and HXT7 from *S. cerevisiae*, XltA from *A. niger*, XylE from *E. coli*, An25 from *N. crassa*; the red rectangles are labeled with conservative sequences KG-XX-LEE. Growth of C-terminal truncation mutant strains in (**B**) solid and (**C**) liquid synthetic medium containing 2% or 0.1% xylose for 5 days. Error bars represent the standard deviation of the biological triplicates.

**Figure 6 ijms-25-08322-f006:**
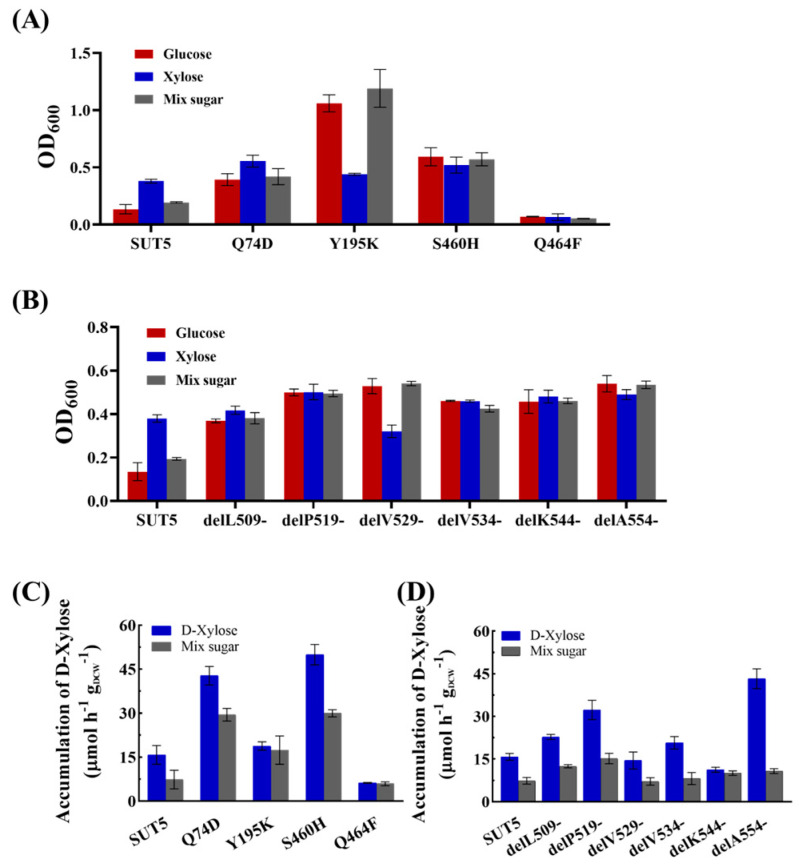
Effects of molecular modification on the inhibitory effect of glucose. Growth of (**A**) point mutation and (**B**) C-terminal truncation mutant strains in 2% glucose, 2% xylose, and mixed-sugar cultures, respectively. Accumulation of intracellular xylose in 2% xylose or 2% mixed sugar (2% xylose and 2% glucose) for (**C**) point mutation and (**D**) C-terminal truncation mutants. Error bars represent the standard deviation of the biological triplicates.

## Data Availability

The datasets generated during and/or analyzed during the current study are available from the corresponding author on reasonable request.

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
