# Peer review of "Molecular Modification Enhances Xylose Uptake by the Sugar Transporter KM_SUT5 of Kluyveromyces marxianus"

_ijms, 2024, doi:10.3390/ijms25158322_

Round 1

Reviewer 1 Report

Comments and Suggestions for Authors

In this manuscript, Luo et al focuses on the sugar transporter gene KM_SUT5 from Kluyveromyces marxianus GX-UN120, highlighting its efficiency in transporting sugars. Through cloning, expressing, and modifying the KM_SUT5 gene, they identify specific mutations that enhance the transporter’s efficiency and provides insights into the structural-functional relationships of sugar transporters. There are several critical areas that require further clarification and refinement before the manuscript is suitable for publication.

1. The figures in this manuscript are too small, making it difficult to see any details.

2. The criteria for selecting the specific amino acid sites for mutation need to be more clearly justified and explained.

3. The mechanistic basis for the enhanced transport efficiency observed in the mutants, particularly the structural changes and their impact on transporter function, is not sufficiently detailed. particularly the role of the C-terminal truncations. How do the truncated versions of KM_SUT5 compare in terms of stability compared to the wild-type transporter?

4. What hypotheses do the authors have about the role of the C-terminal region in transporter function, and how do the truncations specifically enhance transport efficiency? Did the truncations affect the overall stability or membrane localization of the transporter?

Author Response

Response to Reviewer 1 Comments

1.Summary

Thank you very much for taking the time to review this manuscript. Please find the detailed responses below and the corrections highlighted in the re-submitted files

  1. Point-by-point response to Comments and Suggestions for Authors

Comments 1. The figures in this manuscript are too small, making it difficult to see any details.

Response 1: Thank you for raising this matter. The figures in the revised version have been resized to enhance readability.

Comments 2. The criteria for selecting the specific amino acid sites for mutation need to be more clearly justified and explained.

Response 2: Thank you for your suggestion. In the process of selecting specific amino acid sites for mutation, we are utilizing the results of molecular docking in point mutation experiments. The determination of amino acid sites for mutation is also based on the binding energy obtained from docking analysis. These details will be clarified in the revised version.

Comments 3. The mechanistic basis for the enhanced transport efficiency observed in the mutants, particularly the structural changes and their impact on transporter function, is not sufficiently detailed. particularly the role of the C-terminal truncations. How do the truncated versions of KM_SUT5 compare in terms of stability compared to the wild-type transporter?

Response 3: Thank you for the valuable suggestion. It is an excellent question, and the inquiry into protein stability warrants further research and discussion, as emphasized in the manuscript's discourse. Moving forward, we should allocate more attention to the state of structural homeostasis.

Comments 4. What hypotheses do the authors have about the role of the C-terminal region in transporter function, and how do the truncations specifically enhance transport efficiency? Did the truncations affect the overall stability or membrane localization of the transporter?

Response 4: Thank you for your valuable suggestions. The C-terminal region was truncated separately to investigate its potential impact on the transporter's functionality, as it was observed during modeling that the remnants in this region exhibited a long irregular line structure distinct from the alpha helix present in the transmembrane region. Although we were uncertain whether this effect was direct or indirect, subsequent studies indicated a possible correlation with the protein's structural stability.

Reviewer 2 Report

Comments and Suggestions for Authors

The manuscript describes how several mutations to a sugar transporter protein improve the xylose uptake of a yeast species. It includes molecular modeling and experimental testing. This may have practical significance in biofuel production. The methodology is sound and the conclusions are supported, but the presentation should be improved:

1. The figures are very small and most of them are difficult or almost impossible to read.

2. The notation of truncation mutants should be changed. "P519" is not an acceptable way to indicate a truncation mutant as it is unclear. Use e.g. "delP519-" or something like that.

3. Tables 1 and 2 contain plasmid and primer information; they are too long, and should be moved to the Supplementary Material.

4. On the other hand, I would advise that Figures S1 and S2 should be moved to the main article. There is also a "Video S1" which I have not found.

5. Presentation of the molecular docking results should be a separate subsection within the Results, and not lumped together with experimental findings. I would advise that the modeled structures of the sugar binding sites of the mutants (with the docked xylose) should be presented as figures, and the changes in the ligand binding relative to the wild-type protein should discussed (e.g. change in hydrogen bonding, etc.). In the Methods section, specify whether the docking was rigid docking or some residues and/or the ligand were treated as flexible.

6. Introduction lines 63-64: "To gain insights into its functional characteristics, we conducted investigations using the hxt null S. cerevisiae strain EBY-XYL." It is not clear what "hxt null S. cerevisiae strain EBY-XYL" is, please define and explain here.

7. Introduction lines 71-83 are just a summary of the whole paper, which is not necessary here. Shorten it to a single sentence or two.

8. Results line 86: "The KM_SUT5 gene was genetically by fusing the GFP gene at its C-terminus". Was genetically what? A verb is missing.

9. Figure 4 legend, line 175: "Comparison of residues bound to xylose in XylE and SUT5". "XylE" appears here for the first time in the paper. What is this and why did you include it? Explain earlier in the manuscript. Also, in other places it is written as "XyLE", which one is correct?

10. Results lines 196-197: "Under conditions of 0.1% xylose, the original strain demonstrated only 60% growth relative to that observed in 2% xylose; however, its growth rate reached up to82%". So is it 60% or 82%? Not clear. Also, a space is missing.

11. The Discussion starts with this sentence: "The substitution of lysine for the phenyl ring at Y195 significantly impacts the glucose transport process of the KM_SUT5 protein." I would advise that the Discussion should start with a more general overview of the findings and their significance, rather than some particular detail.

I suggest that the manuscript be accepted after minor revisions.

Comments on the Quality of English Language

The English is mostly fine, some minor corrections are needed (missing verb, etc.)

Author Response

Response to Reviewer 2 Comments

1.Summary

Thank you very much for taking the time to review this manuscript. Please find the detailed responses below and the corrections highlighted in the re-submitted files

  1. Point-by-point response to Comments and Suggestions for Authors

Comments 1. The figures are very small and most of them are difficult or almost impossible to read.

Response 1: Thank you for raising this matter. The figures in the revised version have been resized to enhance readability.

Comments 2. The notation of truncation mutants should be changed. "P519" is not an acceptable way to indicate a truncation mutant as it is unclear. Use e.g. "delP519-" or something like that.

Response 2: The revised version of our nomenclature regarding truncating mutations has been updated, thanks to your valuable suggestion.

Comments 3. Tables 1 and 2 contain plasmid and primer information; they are too long, and should be moved to the Supplementary Material.

Response 3: Revised based on your valuable suggestions, we have included Tables 1 and 2 in the updated version of the supplementary material.

Comments 4. On the other hand, I would advise that Figures S1 and S2 should be moved to the main article. There is also a "Video S1" which I have not found.

Response 4: Thank you for your suggestions. Regarding Video S1, there was a misunderstanding. We did not delete any parts of the template regarding supplementary material during the writing of the manuscript, and the supplementary material of this manuscript does not include video files. We will also correct this mistake in the revised version.

Comments 5. Presentation of the molecular docking results should be a separate subsection within the Results, and not lumped together with experimental findings. I would advise that the modeled structures of the sugar binding sites of the mutants (with the docked xylose) should be presented as figures, and the changes in the ligand binding relative to the wild-type protein should discussed (e.g. change in hydrogen bonding, etc.). In the Methods section, specify whether the docking was rigid docking or some residues and/or the ligand were treated as flexible.

Response 5: Thank you for your suggestion. In the revised version, we will provide a more systematic description of molecular docking. The docking of mutant and wild-type proteins to ligands is solely conducted for the purpose of evaluating docking energy, which will be explained in detail in the Supplementary Material. Furthermore, we will elaborate on the specific methodology employed for docking in the Methods section.

Comments 6. Introduction lines 63-64: "To gain insights into its functional characteristics, we conducted investigations using the hxt null S. cerevisiae strain EBY-XYL." It is not clear what "hxt null S. cerevisiae strain EBY-XYL" is, please define and explain here.

Response 6: The suggestion you provided is highly appreciated, and we found that the explanation for hxt null was not sufficiently specific. Therefore, in our revised version, we will include a more detailed explanation at line 45 where the term is first introduced.

Comments 7. Introduction lines 71-83 are just a summary of the whole paper, which is not necessary here. Shorten it to a single sentence or two.

Response 7: The section has been succinctly summarized, thanks to your valuable suggestion.

Comments 8. Results line 86: "The KM_SUT5 gene was genetically by fusing the GFP gene at its C-terminus". Was genetically what? A verb is missing.

Response 8: Thank you for your valuable comment, there is indeed a verb missing in this sentence and we have corrected it in the revised version.

Comments 9. Figure 4 legend, line 175: "Comparison of residues bound to xylose in XylE and SUT5". "XylE" appears here for the first time in the paper. What is this and why did you include it? Explain earlier in the manuscript. Also, in other places it is written as "XyLE", which one is correct?

Response 9: The revised version of the manuscript now incorporates your valuable feedback. The term "XylE" has been standardized throughout, with a corresponding note included in the body of the text.

Comments 10. Results lines 196-197: "Under conditions of 0.1% xylose, the original strain demonstrated only 60% growth relative to that observed in 2% xylose; however, its growth rate reached up to82%". So is it 60% or 82%? Not clear. Also, a space is missing.

Response 10: Thank you for your valuable suggestion. We realized that there was a problem with this sentence and changed it to "The original strain exhibited a growth rate of only 60% under conditions of 0.1% xylose, whereas its growth rate increased to 82% in the presence of 2% xylose." in the revised version.

Comments 11. The Discussion starts with this sentence: "The substitution of lysine for the phenyl ring at Y195 significantly impacts the glucose transport process of the KM_SUT5 protein." I would advise that the Discussion should start with a more general overview of the findings and their significance, rather than some particular detail.

Response 11: Thank you for your feedback. We acknowledge that the direct statement may appear abrupt, so we have included the following sentence in the revised version: "The evident impact of molecular modifications on the transport capacity of sugar transporter proteins." This addition aims to enhance the logical flow of the paragraph.

Round 2

Reviewer 1 Report

Comments and Suggestions for Authors

The authors have have addressed all my comments appropriately. I would recommend the manuscript is ready for publication.